# Tissue Engineered Transcatheter Pulmonary Valved Stent Implantation: Current State and Future Prospect

**DOI:** 10.3390/ijms23020723

**Published:** 2022-01-10

**Authors:** Xiling Zhang, Thomas Puehler, Jette Seiler, Stanislav N. Gorb, Janarthanan Sathananthan, Stephanie Sellers, Assad Haneya, Jan-Hinnerk Hansen, Anselm Uebing, Oliver J. Müller, Derk Frank, Georg Lutter

**Affiliations:** 1Department of Cardiovascular Surgery, University Hospital Schleswig-Holstein (UKSH), D-24105 Kiel, Germany; zhang_xiling@outlook.com (Z.X.); thomas.puehler@uksh.de (T.P.); jette.seiler@uksh.de (J.S.); assad.haneya@uksh.de (A.H.); 2DZHK (German Centre for Cardiovascular Research), Partner Site Hamburg/Kiel/Lübeck, D-20251 Hamburg, Germany; anselm.uebing@uksh.de (A.U.); oliver.mueller@uksh.de (O.J.M.); derk.frank@uksh.de (D.F.); 3Department of Functional Morphology and Biomechanics, Zoological Institute, Christian-Albrecht University of Kiel, D-24105 Kiel, Germany; sgorb@zoologie.uni-kiel.de; 4Department of Centre for Heart Valve Innovation, St Paul’s Hospital, University of British Columbia, Vancouver, BC V5K 0A1, Canada; jsathananthan@providencehealth.bc.ca (J.S.); SSellers@providencehealth.bc.ca (S.S.); 5Department of Congenital Heart Disease and Pediatric Cardiology, University Hospital Schleswig-Holstein, Campus Kiel, D-24105 Kiel, Germany; JanHinnerk.Hansen@uksh.de; 6Department of Cardiology and Angiology, University Hospital Schleswig-Holstein (UKSH), D-24105 Kiel, Germany

**Keywords:** tissue engineering, heart valve replacement, pulmonary, transcatheter, percutaneous, biodegradable, nitinol, stents, congenital heart disease, decellularization, recellularization

## Abstract

Patients with the complex congenital heart disease (CHD) are usually associated with right ventricular outflow tract dysfunction and typically require multiple surgical interventions during their lives to relieve the right ventricular outflow tract abnormality. Transcatheter pulmonary valve replacement was used as a non-surgical, less invasive alternative treatment for right ventricular outflow tract dysfunction and has been rapidly developing over the past years. Despite the current favorable results of transcatheter pulmonary valve replacement, many patients eligible for pulmonary valve replacement are still not candidates for transcatheter pulmonary valve replacement. Therefore, one of the significant future challenges is to expand transcatheter pulmonary valve replacement to a broader patient population. This review describes the limitations and problems of existing techniques and focuses on decellularized tissue engineering for pulmonary valve stenting.

## 1. Introduction

The incidence of congenital heart disease (CHD) is approximately 9‰ in newborns [1]. Approximately 20% of newborns with CHD have malformations of the pulmonary valve or right ventricular outflow tract (RVOT), such as tetralogy of Fallot, one common arterial trunk, or pulmonary atresia [2]. In this patient group, surgical correction in the first month of life improves patient prognosis [3]. Surgical strategies for RVOT reconstruction include transannular patch enlargement, bioprosthetic valve placement, and valved conduits [4]. The tolerability of surgical intervention of the RVOT for complex CHD depends on the patient’s age and the type of tissue material used [4,5].

However, multiple complications such as pulmonary regurgitation, self-growth, external ductal stenosis, valvular stenosis, ductal distortion, sternal compression, intimal hyperplasia, external ductal calcification, and deformation of the aneurysm lead to progressive RVOT dysfunction in these patients [4,6].

Therefore, repeated surgical interventions are required throughout their lives. Despite the low mortality rate of surgical interventions, complications are significantly higher, especially in patients undergoing repeated procedures [7,8].

In this clinical context, transcatheter pulmonary valve replacement (TPVR) has emerged as a non-surgical, less invasive alternative treatment for RVOT dysfunction.

In 2000, Bonhoeffer et al. [9] successfully performed the first TPVR using a bare-metal platinum alloy stent and a manually sewn flap bovine jugular vein [2,10]. In the decade since then, catheter-based valve implantation has evolved rapidly. As a result of TPVR, the interval between surgical replacements has approximately doubled [11]. Although the results are encouraging, the heterogeneity of this patient population and the diversity of morphology, size, and hemodynamics of the implantation site allow for TPVR in only about 15% of patients, which means that surgical pulmonary valve replacement is still necessary for 85% of patients, limiting the applicability of TPVR [12].

In addition, valve implantation is not without complications. The prosthetic valves currently used in clinical practice have significant deficits. Mechanical valve replacements require lifelong anticoagulation therapy, with the attendant risk of bleeding and thrombo-embolism [13], while biological valves are poorly durable, with an average useful life of only about 7 to a maximum of 15 years, due to changes such as calcification and decay in the distant future [14,15]. Moreover, since the allograft or homograft may never grow, it was a pitfall for younger patients, meaning they may have to undergo multiple surgeries [16]. Despite the use of transcatheter valves, interventions were required every 2.6 years (median) [11].

Tissue engineering is a multidisciplinary field that aims to develop biological tissues that can be used in the clinical treatment of diseases. Tissue-engineered products have proven to be effective in different applications, such as burn treatment or drug screening. The success of this approach in other medical fields has established the foundation for its application in heart valves [17,18,19,20,21,22]. A new alternative therapy to replace faulty valve grafts is emerging—tissue-engineered heart valves. These limitations are being overcome by tissue-engineered heart valves that are living, non-cytotoxic, and mechanically analogous heart valve replacements, and can grow and remodel with the patient. However, living valves still have insurmountable problems, including the choice of living material, the complexity of in vitro culture, and how the valves are stored. All of these issues have hindered the commercialization of tissue-engineered heart valve therapies.

This review discusses the applications and limitations of TPVR and the practice and prospects of tissue-engineered valved stents.

## 2. Transcatheter Pulmonary Valve Replacement in Large Right Ventricular Outflow Tracts

TPVR currently has a high success rate with favorable safety and prognosis. A 96.2% success rate was reported in a previous meta-analysis [23]. Data from US clinical trial data, the freedom from 5-year reintervention and reinsertion rates were reported to be were 76 ± 4% and 92 ± 3%, respectively [24]. However, the current indications for TPVR are limited to postsurgical RVOT dysfunction with an enlarged ductal internal diameter between 16–24 mm (Melody valve, Figure 1A) and 21–27 mm (SAPIEN valve, Figure 1B) [16,25,26]. It is estimated that only 15% of patients with CHD and RVOT dysfunction meet such stringent criteria [12,27]. Most patients who would benefit from TPVR are those with non-indicated applications, such as native RVOT or RVOT enlarged by transannular patch, bioprosthetic valves, or small internal diameter (<16 mm) of external ducts.

Boshoff et al. [27] reported 23 cases of TPVR applied non-indicatively, including 8 cases of enlarged RVOTs with transannular patches without external conduits, 2 cases of pulmonary valve stenosis, and 13 cases of external conduits with small internal diameters. At a mean follow-up time of 1.2 years, the peak RVOT pressure difference was significantly lower, and no more than mild pulmonary regurgitation was present. Two patients were re-intervened for the development of restenosis. No vascular complications, stent disruption, or valve migration occurred during follow-up. Meadow et al. [28] found a solution to the right ventricle-to-pulmonary artery pressure difference, successfully following 31 patients with TPVR for RVOT dysfunction on their own or without an external conduit. Similar encouraging results were observed in cases of percutaneous “valve in valve” after biologic valve dysfunction. A total of 104 cases of TPVR after biologic valve dysfunction were reported by Gillespie et al. [29] At a median follow-up of 1 year, four cases of restenosis with no more than mild pulmonary regurgitation were observed, and at follow-up, stent disruption was observed in two cases, none of which required further intervention. Overall, the 2-year reintervention-free rate was >90%, and no operation-related deaths occurred.

There are clearly increased complications associated with non-indicated applications compared to strict adherence to indications as well as issues related to liability and ethics for non-indication applications. However, situations exist in which TPVR is clinically indicated in healthcare settings based on published data or standard practice, but it remains non-indicated if applied by regulatory agency regulations. In such cases, patients may be deprived of effective therapeutic measures because of strictly limited indications. As TPVR technology evolves, more and more large clinical studies on the use of TPVR in small inner diameter, non-tubularly placed RVOT cases will be conducted, expanding the indications for TPVR in the near future.

A range of innovative techniques are emerging to expand the population for TPVR, particularly in patients with dilated RVOTs.

In 2010, Schievano et al. [30] reported a case of successful transcatheter self-expanding pulmonary valve expansion (Figure 2). This device was later called the Native Outflow Tract device (Medtronic, Figure 1C). It has an hourglass contour line with relatively large internal diameters at the ends and relatively small internal diameters in the middle. The self-expanding properties of the nitinol stent could, in theory, improve valve stability in the presence of different RVOT morphologies. At 6-month follow-up, no stent fracture and only trivial paravalvular leakages were observed.

The Venous P valve stent (Shanghai, China,Figure 1D) is also a novel self-expanding percutaneous pulmonary device consisting of a three-lobed porcine pulmonary valve and a surrounding nickel-titanium metal stent frame delivered by a 14- or 22-F sheath, with valve sizes ranging from 20 to 32 mm. Small clinical studies have confirmed the safety and efficacy of the Venous pulmonary valve stent [31,32]. Given the ability of this device to fit the enlarged RVOT via a transannular patch, this device will play an important role in widening the indications for TPVR in the future if additional clinical studies confirm these positive results.

Similarly, the Sapien XT (Edwards, Figure 1E) 29 mm size valve can be used as an alternative treatment for larger inner diameter RVOTs; although, this valve has not been explicitly studied in the pulmonary artery position. A percutaneous pulmonary valved stent designed to reduce the internal diameter of the RVOT has been successfully placed in the RVOT of sheep, but it is still in preclinical trials [33,34]. Other advanced techniques have been successfully used in patients with disseminated, complex RVOT geometry, including placement of Melody transcatheter pulmonary valves in both pulmonary arteries [35], and the pulmonary artery “jailing” technique in which a folded bare metal stent is bifurcated from the pulmonary artery into the RVOT to distend it as an anchor for TPVR [36].

However, there are still some problems to overcome, such as tricuspid valve injury, paravalvular leakages, stent migration, arrhythmia, stent fracture, etc. [37]. The stent grafts for patients with large native or patched RVOT diameters (>30 mm) cannot be addressed solely by the transcatheter aortic valve stents currently available on the market. An innovative transcatheter stent must be developed to remove surgical limitations.

### 2.1. Development of Tissue-Engineered Pulmonary Valves

While pulmonary valve replacement can be lifesaving in pediatric and adolescent patients, there is a critical problem: the graft cannot grow. This means that such patients may have to undergo multiple surgeries, increasing the risk of complications and even death. To address this problem, tissue-engineered valves can create a living heart valve with excellent self-repair and reconstruction capabilities that can overcome the various disadvantages of current prostheses. Tissue engineering can enhance the ability of damaged, malformed, or diseased valve tissue to heal itself by guiding the reconstruction of a native extracellular matrix (ECM) bionic microenvironment with appropriate biomechanical properties, as well as providing the necessary bionic physical and biological stimuli. Synthetic resorbable valves with bionic function not only transport cells but also provide mechanical support and bio-stimulation to promote valve regeneration, thus meeting a variety of clinical needs [38]. The construction of novel valve materials by means of tissue engineering is expected to overcome the shortcomings of current valve materials. The tissue-engineered heart valve (TEHV) is constructed by using a combination of porous cell scaffolds, seed cells, and bioactive factors with the ultimate goals of: (1) providing excellent hemodynamics without anticoagulation therapy, (2) promoting tissue remodeling and preventing valve degradation, and (3) having regenerative characteristics to avoid reoperation. TEHV cellular scaffolds mainly include natural material stents (such as decellularized tissue or biomaterials) versus synthetic material stents (degradable polymers).

### 2.2. Decellularized Tissue-Engineered Heart Valve

Decellularization is removing cells (including the nucleus) from the ECM of biological tissue. After cell removal, the remaining ECM provides a cellular scaffold with voids, which not only retains the complex geometry of natural tissues but also consists of natural components that promote cell migration and differentiation, resulting in constructive remodeling. In addition, decellularized valves do not require complete degradation and can maintain their own anisotropic mechanical properties. To date, decellularized heart valves (DHVs) are more favorable than synthetic-material valves for clinical applications, and implantation has been accomplished in animals and humans as valve replacements.

Although decellularized tissues have been used clinically for many years, it was not until 2011 that generally accepted quantifiable minimum criteria for moderate decellularization was generated [39]: in samples with (1) double-stranded DNA (dsDNA) content < 50 ng/mg ECM dry weight; (2) DNA fragment length < 200 bp; and (3) no visible nuclei in sections stained with 4′,6-diamidino-2-phenylindole (DAPI) or hematoxylin-eosin (H&E) stained tissue sections without visible nuclei.

These criteria list the essential indicators to be met for clinically applicable decellularized tissues because incomplete decellularization can affect the human immune response by affecting macrophage polarization and inhibiting constructive matrix remodeling. Although decellularization does preserve the natural tissue geometry, inappropriate decellularization methods can negatively affect the natural three-dimensional ultrastructure of the matrix, the surface topology, and the composition of matrix proteins.

### 2.3. Seed Cells and Recellularization of Tissue-Engineered Valves

The body’s heart valves consist of mainly interstitial valve cells and endothelial cells. Endothelial cells cover the valve leaflet surface and serve as an essential protective barrier between the interstitial cells and the extracellular matrix, while endothelial cells also regulate the coagulation system. The current biological valves in clinical use are pericardial membranes prepared using glutaraldehyde cross-linking. The pericardium itself has no endothelial cells, and the endothelialization, of biologic valves can effectively delay or prevent the decay and calcification of biologic valves, which may fundamentally solve the problem of biologic valve durability, and effort is directed to develop new biologic valves with very important clinical significance [40].

There are two ways of achieving biological valve re-endothelialization [41]: in vitro and in vivo. For in vitro re-endothelialization, host endothelial cells are cultured in vitro and grown on the surface of the bioprosthetic valve. Dohmen et al. [42] isolated, cultured, expanded, and seeded human transcatheter endothelial cells on human post-decellularized pulmonary valves with good results after surgery. In vitro re-endocytosis faces many difficulties and challenges, including the source of seeded cells, efficiently achieving an expansion of seeded cells, and possible immune rejection. In vivo re-endothelialization, or in situ recellularization, refers to the implantation of decellularized biologic valve scaffold tissue directly into the body, where the host’s own endothelial cells grow and crawl. Elkins et al. [43] implanted decellularized sheep pulmonary artery valves into young sheep for 6 months and found good valve recellularization. The mechanisms of in vivo endothelialization identified so far include: (1) crawling growth of endothelial cells from the injured portion; and (2) deposition of seed cells onto the endothelial surface using blood circulation, also known as sedimentation healing. In vivo reendothelialization/in situ recellularizations may be a more suitable strategy for clinical application than in vitro recellularization.

### 2.4. Effects of Seed Cells on Recellularization

The recellularization process is influenced by various factors such as the characteristics of the seed cells and the culture environment, the nature of the scaffold material itself, and the cell–scaffold material interaction. Seed cells can generally be divided into adult cells (primary cells) and stem cells (or precursor cells that differentiate in a specific direction).

Earlier, a variety of adult cells such as endothelial cells, VIC, and myofibroblasts were used for the study of DHVs recellularization. The use of autologous endothelial cells (of vascular or stem cell origin) grown in vitro in DHVs results in a better endothelial cell layer (vWF+) on the valve surface, which provides some protection to the valve tissue [44,45,46,47]. However, in the absence of mesenchymal cells, it is challenging to keep the formed endothelial cell layer stable in the presence of hemodynamic shock from physiological conditions. Other studies have attempted to implant mesenchymal type cells (cardiac mesenchymal cells) in DHVs, or endothelial and mesenchymal type cells (myofibroblasts) together [48,49,50], and have shown that some degree of endothelialization and infiltration of mesenchymal cells (αSMA+, VIM+) can be achieved; although, there is a disparity in cell volume and cell distribution with natural valves.

In recent years, with the increasing maturity of stem cell technology, many studies have started to use stem cells in attempts to recellularize DHVs. The potential for multidirectional differentiation and the ability of self-renewal of stem cells make them theoretically advantageous as seed cells, and the commonly used ones are mesenchymal stem cells (MSC), endothelial progenitor cells, embryonic stem cells, and induced multifunctional stem cells. MSCs are derived from bone marrow or adipose tissues and are most widely used today. They can transform to endothelial cell phenotypes in the presence of vascular endothelial growth factor (VEGF) and under conditions such as higher fluid shear [51,52], and also have mesenchymal cell phenotype transformation potential [53].

Vincentelli et al. [54] injected receptor-derived MSCs into the mesenchyme of porcine DHVs and then implanted them into sheep pulmonary valve sites, which resulted in the formation of an intact endothelial layer and a distribution of mesenchymal cells similar to that of natural pulmonary valves after 4 months. Endothelial progenitor cells are mainly obtained from peripheral blood or umbilical cord blood and differentiate mainly to endothelial cells during maturation, but can also exhibit a mesenchymal cell phenotype under certain conditions [55]. Embryonic stem cells are derived from embryonic tissues and induced multifunctional stem cells are derived from somatic cells. Both have high differentiation potential and can be induced to transition into endothelial or mesenchymal cell types by a suitable environment, providing favorable environmental conditions for the full recellularization of DHVs.

In the process of decellularization, the ECM is inevitably destroyed. Therefore, to improve the ability of seed cells to adhere, proliferate, migrate, and differentiate, bioactive factors such as cytokines, antibodies, and peptides can be applied to modify and promote endothelialization. The main growth factor commonly used is VEGF [56]. VEGF is associated with the recruitment of endothelial cells or endothelial progenitor cells in the bloodstream. VEGF-based surface modification strategies can improve the endothelialization of biological heart valves. Hopkins et al. [57] used bioengineered (decellularized with collagen conditioning treatments) human and baboon heart valve scaffolds implanted in baboons, and the treated valves were less immunogenic, had a less inflammatory response, and were well recellularized with good valve function. Jordan et al. [15] combined CD133+ cells isolated from peripheral blood with the surface of porcine DHVs and achieved complete endothelialization 1 month after implantation into sheep, and mesenchymal cells, matrix metalloproteinases, and collagen were significantly higher after 3 months than in both the DHVs group and the DHVs group implanted with autologous endothelial progenitor cells in vitro. Our group [58] also demonstrated that bone marrow-derived CD133+ cells on DHVs had better results concerning possible calcification, inflammation, and transvalvular gradients compared to autologous carotid artery cells on DHVs in vivo.

### 2.5. Effects of Bioreactors on Recellularization

Heart valves, open and close approximately 100,000 times per day with each heartbeat in an in vivo blood flow environment. Static conditions in culture in vitro are far removed from the physiological environment in vivo. Many studies have attempted to simulate the dynamic environment in vivo to achieve better recellularization. Bioreactors can simulate physiological conditions of pressure pulses and blood flow environment in vitro to pre-adapt valve tissue to the complex environmental changes in vivo before implantation and to provide more favorable conditions for seed cell proliferation and differentiation.

Simulating the in vivo blood flow environment under hydrodynamic stimulation conditions can improve the success rate of re-endothelialization of stent materials. Lichtenberg et al. [59] found that complete endothelialization of DHVs could be achieved by providing hydrodynamic stimulation through a bioreactor and stepping up the flow rate of the solution in the reactor to physiological conditions, whereas increasing the flow rate too rapidly to physiological conditions resulted in the destruction of endothelial cells implanted on the surface of the scaffold material, which may explain the shedding of implant cells when tissue-engineered valves formed under static in vitro culture enter the body and are exposed to the physiological mechanics [60]. They further re-endothelialized DHVs using young sheep autologous endothelial cells in a bioreactor, and after 3 months of implantation into young sheep pulmonary valve sites, the endothelial coverage of the valve was increased and thrombus formation on the valve surface was reduced [61]. A similar approach was used for the implantation of sheep aortic valves, which showed no significant dysfunction after 3 months, with intact endothelial coverage and no signs of inflammation, whereas the cryopreserved implanted valves showed significant signs of calcification and decay [62].

A dynamic culture environment that mimics physiological conditions may also increase infiltration of mesenchymal type cells and facilitate the preservation of valve cell function after implantation. Schenke-Layland et al. [63] incubated myo-fibroblasts with DHVs under static conditions for 2 d, then transferred them to a bioreactor for 9 d or 16 d before implanting endothelial cells in a static environment, which resulted in a better recellularization of the valve compared with static culture alone, with better coverage of the endothelial layer and significant infiltration of mesenchymal cells.

Kajbafzadeh et al. [64] grew autologous MSCs in decellularized sheep aortic valves under pulsed conditions in a bioreactor, cultured them, and implanted them in the thoracic descending aortic position in sheep, eight recipients survived up to 19 months with no significant endothelial tears on CT at 18 months, and pathological sections showed that the valves reached a level of α SMA+ cellularity closer to that of natural valves. Cebotari et al. [65] grew single nucleated cells isolated from human peripheral blood in a bioreactor in decellularized human pulmonary valves. After 21 d of culture, endothelial cell-specific phenotypes were seen on the surface cells, which were then implanted in the pulmonary valve sites of two children with preexisting heart disease. During 3.5 years of follow-up, it was observed that the implanted pulmonary valves maintained relatively good function and showed growth without significant failure.

### 2.6. In Vivo In Situ Recellularization Procedure

In contrast to in vitro recellularization, many research teams have attempted to implant DHVs directly into the body, hoping that it will guide specific cell aggregation, adhesion, proliferation, and differentiation in vivo to achieve in situ recellularization, which is also called “guided tissue regeneration” [66]. Compared to in vitro recellularization, this method recruits and differentiates cells in vivo without the need for autologous cell isolation, implantation, and culture processes, and the in vitro treatment process is relatively simpler and shorter. Some studies have found that in vivo implantation in mice results in more complete in situ recellularization [67], but most implanted valves in pigs or sheep show only endothelial coverage or infiltration of inflammatory cells [66,68,69,70,71,72,73,74], suggesting that it may be more difficult to achieve complete recellularization directly in vivo in larger animals. Rabbit aortic valves implanted in dogs showed rapid failure [75], whereas porcine-DHVs implants in dogs or sheep developed better endothelialization [76,77], suggesting that small animal valves implanted in large animals may be less likely to achieve recellularization and less durable.

### 2.7. Tissue-Engineered Pulmonary Valves in the Clinic

To date, young patients including pediatric patients undergoing pulmonary valve replacement continue to face the dilemma that neither mechanical valves nor xenobiotic valves are ideal. Therefore, the homogeneous allogeneic pulmonary valve remains the best option currently: cryopreserved, homogeneous valves have good hemodynamic performance, low incidence of thromboembolism and infection, and better durability than other biologic replacements. A concomitant problem, however, is the degradation of homograft valves over time, which is particularly evident in children and young adults [16].

The results of animal trials with decellularized valves have been encouraging, with animals surviving up to 9 months after surgery, but their clinical trial results have been mixed: in clinical practice, the majority of decellularized valves used are xenografts rather than homografts due to the scarcity of human tissue, and clinical practice with xenograft decellularized valves has not been encouraging. The SynerGraft^®^ valve from CryoLife, a decellularize valve of porcine origin, showed good initial results in adults, but has triggered a severe immune response in pediatric patients [78].

Another product, Matrix P^®^ and Matrix P plus, required reoperation for valve failure in 52% of children after implantation, and the implanted valve leaflets showed significant thickening with fibrosis, and severe inflammation without significant re-endothelialization [79,80]. Overall, although advances in the decellularization process or better pre-implantation conditioning may improve outcomes, the clinical performance of decellularized allogeneic valves is not superior to that of standard cryopreserved homogeneous valves.

On the other hand, human homogeneous decellularized valves have been more successful in clinical practice. Haverich et al. [81] conducted a decade-long study of decellularized pulmonary homografts (DPHs) valved stents. The results showed that the mid-term results of DPHs for PVR confirm earlier results of reduced re-operation rates compared with cryopreserved pulmonary homografts (CHs) and bovine jugular vein (BJVs) conduits, and growth potential is one of the unique features of DHPs.

Preliminary results from a recent European multicenter prospective trial show that DPH has an appreciable safety and efficacy profile with good hemodynamics over a period of up to 3 years [82]. Evidence from multiple centers also indicates that decellularized allografts are consistent with conventionally cryopreserved allografts in a comparison of early and mid-term outcomes. Moreover, the explantation and degeneration rates are lower than those of conventional cryopreserved allografts [83,84,85].

In addition, data show that the CryoValve^®^ SG valve, a human allogeneic decellularized valve developed by CryoLife, did not require reoperation in 93% of patients at 10 years, and was free of valve dysfunction in 85% of patients at 5 years after implantation and 75% of patients at 10 years after implantation [86]. Compared with the gold standard frozen (cryopreserved) human-derived valve (homografts), the CryoValve^®^ SG valve was less likely to develop valve dysfunction and less likely to require re-intervention. However, the mean follow-up was only 5 years in this study, and more extended observation is still needed [86]. Table 1 demonstrates the clinical application of decellulized homogeneous pulmonary valves.

The primary advantage of decellularized valves over the gold standard cryopreserved human-derived valves (homografts) is the reduced antigenicity. This is because endothelial cells that survive cryopreservation can still express HLA antigens intact and undergo immune reactions that cause valve failure [87]. However, to date, complete autologous recellularization has not been achieved in implanted DHVs. Autologous recellularization of decellularized valves is limited to the valve wall, whereas on the leaflet surface, only the recellularization of endothelial cells has been observed. Although this has been significantly superior to cryopreserved valves (because of the severe degradation and leukocyte infiltration throughout the valve after frozen valve implantation), the problem that recellularization of decellularized valves is limited to the valve wall still limits their clinical application because the valve leaflet is the main site of dysfunction after cryopreserved valve implantation.

Without the ability to reconstitute living cell communities within the leaflet mesenchyme that can repair and remodel the ECM, DHVs will suffer the same fate as cryopreserved valves, i.e., gradual valve degradation after implantation. In addition, DHVs require human or animal tissue to manufacture, which is in limited supply and requires cryopreservation, yet freeze-drying destroys the structure of the ECM in the valve and leads to inactivation of biomolecules.

In addition, we should consider the cytotoxicity of decellularized tissue-engineered valves. Detergents commonly used are sodium deoxycholate (SDC), sodium dodecyl sulfate (SDS), and ethylenediaminetetraacetic acid (EDTA). Even after rinsing, detergent residue will remain. After surface seeding, Rieder et al. found that SDS-decellularized xenomatrices had a toxic effect on endothelial and myofibroblast cells [46]. Nevertheless, residue concentration (<50 mg/L) in the wash solution has not affected the receptiveness of decellularized matrices to human endothelial cells. Therefore, detergent concentrations less than 50 mg/L in the washing solution may be used as an indicator for the production of non-toxic tissues. Alternatively, a combination of detergents at medium concentrations can achieve good decellularization and can also be rinsed off more easily [88]. Decellularized TEHV needs to have long-term storability. Long-term storage may result in increased decellularized residues due to degradation of the ECM. The goal of future research should be to find optimal decellularization protocols and storage strategies since decellularization protocols and storage strategies still differ.

Despite the many advantages of allograft valves, both decellularized and conventional cryopreserved allograft valves have associated problems. In addition to the scarcity of raw materials, calcification occurs. This also dramatically reduces its availability. Currently, two types of materials are being studied, biological-based materials and synthetic-based materials. Decellularized valves made from collagen, elastin, fibrin, sodium alginate, chitosan, and other biomaterials have advantages in cell adhesion, migration, proliferation, and differentiation. However, their mechanical properties are fragile. Many polymeric heart valves are already in use. In fact, pulmonary monocusp valve prostheses made of polytetrafluoroethylene have been used for more than 20 years for the surgical treatment of RVOT abnormalities due to calcification or abnormal function in young patients with good results [95]. However, these materials were found to degrade and thrombose at an early stage [96]. Lutter et al. [97] described a novel low-profile self-expanding nitinol stent carrying a dip-coated pulmonary heart valve prosthesis made of polycarbonate urethane (PCU) and demonstrated promising results in animal experiments (Figure 3). PCU has been shown to have good biocompatibility, durability, and resistance to thromboembolism. It is expected to be an alternative to allograft pulmonary valves. Seed cells were also successfully colonized on polymeric heart valves. Aleksieva et al. [98] successfully seeded saphenous vein-derived endothelial cells and fibroblasts on polyurethane scaffolds. CD133+ cells were also successfully colonized on PCU [58]. Combining synthetic scaffolds with biological scaffolds is also being explored; for example, chitosan-modified polycaprolactone porous scaffolds can improve fibroblast adhesion on TEHV [99]. The use of synthetic heart valves in clinical applications, however, has not yet been reported.

## 3. Tissue-Engineered Transcatheter Pulmonary Valved Stent Implantation

In patients with RVOT dysfunction, TPVR offers a minimally invasive treatment option that allows such patients to avoid repeat surgery. In contrast, the bovine jugular vein valve currently used in TPVR is a glutaraldehyde fixed heterogeneous valve that is prone to degeneration. Tissue-engineered valves offer a way to overcome the limitations of current heart valve replacements. Tissue-engineered valves are currently being used successfully in the clinic with good results. However, the current approach to tissue-engineered valve implantation still uses open-heart surgery rather than transcatheter minimally invasive surgery. There are still some tissue-engineered valve grafts in studies with stenosis and/or regurgitation, which are mainly caused by distal anastomotic stenosis [81]. Furthermore, valve regurgitation can occur when implantation is poor because decellularized grafts are soft. The growth potential is a unique feature of decellularized pulmonary valve grafts, but it depends on the recellularization of the graft. However, when the graft is in the setting of severe scarring, rapid and complete recellularization seems unlikely. Therefore, how to combine TPVR with tissue-engineered valves should be a focus of future research. While avoiding repeat surgery and overcoming the disadvantages of existing xenobiotic valves, Lutter’s group [100,101] successfully performed TPVR in a sheep model using an autologous tissue-engineered valve scaffold without regurgitation after 4 weeks, and angiography and echocardiography showed good opening and closing of the implanted heart valves. However, all animals developed arrhythmias (Figure 4).

There are several issues that still need to be addressed for transcatheter tissue-engineered pulmonary valve stenting. The first is the anchoring of the valve stent. It is essential to deliver the stent to the exact position and not displace it. A precise 3D structural design allows for better alignment of the stent and better adaptation to the RVOT and pulmonary valve annulus region at implantation.

The second is the choice of the stent material. Bare metal stents are unable to follow the growth of the child and have a limited ability to expand, which means that surgical re-intervention is required. Biodegradable stents have become an alternative to bare metal stents, with the most significant advantage of degradation leaving only endothelialized natural vessels with the potential for further growth. Teams worldwide have been working on designing biodegradable stents for children with CHD [102]. Illusicor stent (Tremedics Medical Devices LLC, Richland Hills, TX, USA) and 480 Biomedical stent (480 Biomedical Inc., Cambridge, MA, USA) have been developed as bioresorbable stents with preclinical experience for use in CHD [102,103,104]. The Zinc bioresorbable stent (ZeBRa stent, Pediastent LLC, Cleveland, OH, USA) is the latest biodegradable stent explicitly designed for CHD.

## 4. Discussion

TPVR is a safe and effective non-surgical treatment strategy for RVOT dysfunction. However, TRVR is still not available in approximately two-thirds of patients due to large RVOT diameters. Devices suitable for large RVOT diameters are currently under investigation and testing (Venus P valve, Medtronic Harmony). The largest transcatheter pulmonary valve prosthesis is the Edwards Sapien XT valve, a transcatheter aortic valve implant. It can be dilated to 31 mm without damaging the Sapien valve or using an Alterra Adaptive Prestent (Edwards Lifesciences, Irvine, CA, USA) [105]. In future work, an innovative valved stent must be developed to address these limitations. Until then, surgical treatment will remain the gold standard.

In addition to this, the issue of valve material deserves our attention. For young patients who need to be operated, multiple surgeries are to be avoided. Therefore, tissue-engineered valves have been studied for more than 20 years as a solution to this problem. DHVs are an essential component of this research. Decellularized pulmonary valve homografts (DPVHs) have been used in the clinic, and the first fDPVHs have been shown to perform with spontaneous recellularization potential [65,81]. fDPVHs have demonstrated superior performance compared to CHs [82]. cDPVHs also showed promising results in terms of immunocompatibility, performance, and durability in the short term [91,106]. However, in an 8–10-year time period, it did not significantly reduce reoperation rates nor did it show better performance than CHs and was comparable to CHs in terms of fibrosis, calcification, and the degree of recellularization [79,92,106,107].

In contrast, it has been shown that fresh DPVHs performed better than cryopreserved DPVHs [106], possibly because the hypothermic environment and decellularization procedure affected the original tissue structure and mechanical properties, thus affecting the endothelialization procedure. In addition, it has been shown that low-dose gamma radiation can severely affect the structural integrity of ECM [108].

With the development of multiphoton-induced autofluorescence and second harmonic generated imaging, it is possible to visualize the alterations of the ECM in frozen heart valve tissue [109]. It is well known that the formation of extracellular ice is a hazard to structural tissues and organs. In 1965, Farrant recommended replacing 60% of cell water with cryoprotectant to prevent freezing at temperatures below −70 °C [110]. As a result, several valve tissue preservation methods have been developed. To promote vitrification, a 55% cryoprotectant formulation can be used to avoid ice formation below the glass transition temperature of the cryopreservation solution [111]. Brockbank et al. reported the use of an 83% cryoprotectant formulation: 83% of cryoprotectant formulation are stored at higher temperatures than 55% cryoprotectant formulation and cut without the risk of ice formation. This method is called ice-free cryopreservation, and the study showed that there was no significant difference in material properties, ECM component integrity, and cell viability of porcine valve tissue stored under 83% cryoprotectant solution at −80 °C compared to −135 °C vitrification cryopreservation [112]. However, cryoprotectants in high concentrations are potentially toxic, so long-term studies are still needed to determine their safety.

While other storage methods such as glutaraldehyde [113], glycerol [114], etc., have been reported, there is no consensus so far on the best storage strategies to ensure the quality of decellularized TEHV.

Therefore, it is essential to investigate cryopreservation techniques and optimize the decellularization and sterilization processes to protect homograft stability and remodeling after this step.

The lack of homografts has prompted researchers to search for new polymeric valve materials. In preclinical studies, polymeric valves have shown promising early function, implantation remodeling, and endothelialization potential [115,116,117]. However, they are still not being used in clinical applications. The reasons for this may be manifold; Firstly, their long-term safety and efficacy have not yet been established. Secondly, the whole process, both technically and logistically, is complex. Finally, differences between donor-to-donor may lead to inconsistent and uncontrolled leaflet thickening and shortening of the final product, resulting in valve insufficiency [118]. Out of these reasons, decellularized heart valves are more favorable than synthetic ones at the time [81,82].

In situ tissue engineering has received much attention, is a simple process compared to in vitro re-cellularization and is designed to promote host cell adhesion and tissue formation. However, in situ tissue engineering is more dependent on the recipient’s regenerative potential. The ideal in situ TEHV tissue should have the ability to selectively control host cell aggregation and adhesion, as well as the ability to directionally differentiate while controlling the onset of degradation until near-natural, functional tissue emerges [119].

## 5. Summary and Prospects

TPVR has emerged as a safe and effective therapy for RVOT dysfunction. The development of tissue engineering has solved already some problems. Nevertheless, there are still some issues that limit its indications and long-term prognosis.

Therefore, new designs of tissue-engineered bioresorbable valved stents and the search for more suitable valve material and, more importantly, combining the two technologies with each other will be the focus in the near future.

## Figures and Tables

**Figure 1 ijms-23-00723-f001:**
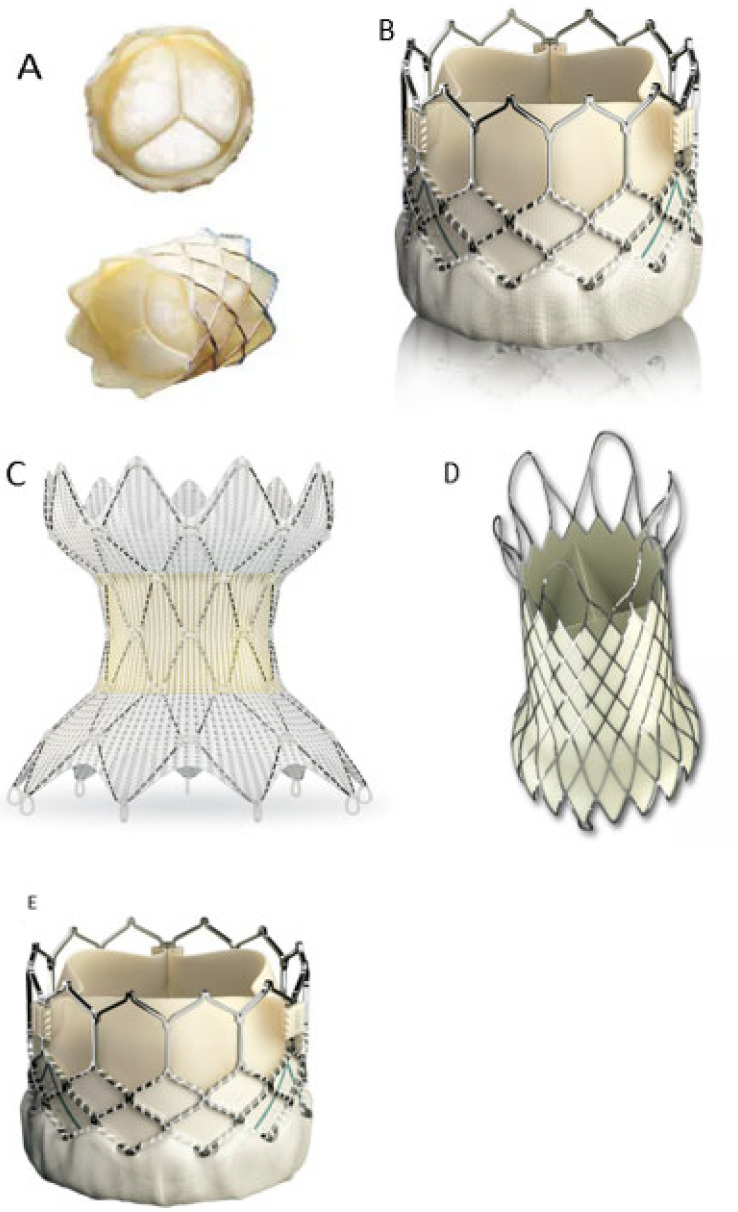
(**A**) Medtronic Melody Pulmonary valve, comprised of a bovine jugular vein valve sutured within a platinum iridium frame; with the permission of reproduction by Medtronic Inc. (**B**) Edwards SAPIEN 3 valve, made of bovine pericardial tissue attached to a balloon-expandable, cobalt-chromium frame for support; with the permission of reproduction by Edwards Lifesciences Inc. (**C**) Medtronic HARMONY Transcatheter Pulmonary Valve, comprised of porcine pericardial tissue valve and self-expanding nitinol frame with polyester cloth covering; with the permission of reproduction by Medtronic Inc. (**D**) VenusP-Valve, consisting of a self-expanding stent made of nitinol with a tri-leaflet porcine pericardial tissue; with the permission of reproduction by Venus Medtech Inc. (**E**) Edwards SAPIEN XT valve, made of bovine pericardial tissue with high radial strength cobalt-chromium frame with low frame height design; with the permission of reproduction by Edwards Lifesciences Inc.

**Figure 2 ijms-23-00723-f002:**
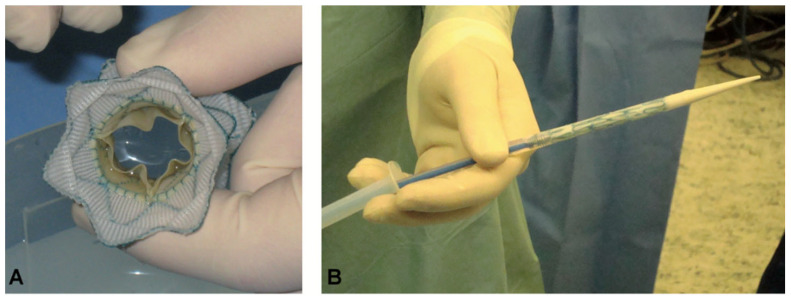
New device: (**A**) An end-on-view of the new device showing the Nitinol stent with graft covering and the open pericardial valve leaflets. (**B**) The delivery system with the device crimped and loaded onto the distal end (right side of picture) [30].

**Figure 3 ijms-23-00723-f003:**
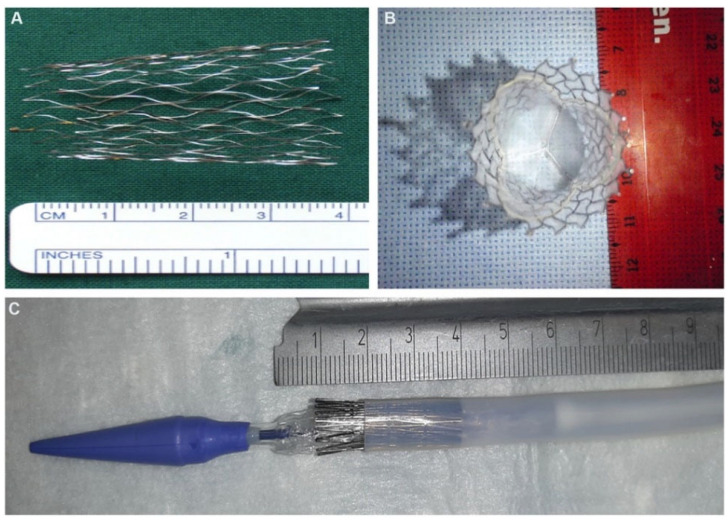
(**A**) Self-expanding nitinol stent, sizes 24 mm (ventricular)/22 mm (middle)/20 mm (pulmonary) with an overall length of 35 mm. (**B**) Top view of tricuspid polycarbonate urethane valved stent. (**C**) The valved stent is crimped in a 14-Fr delivery catheter with a smooth silicone structure at the proximal tip [97].

**Figure 4 ijms-23-00723-f004:**
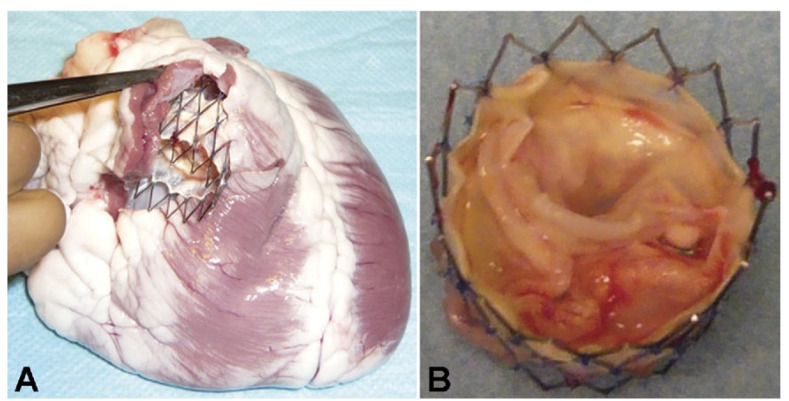
(**A**,**B**) Gross morphology of tissue-engineered pulmonary valved stent [100,101].

**Table 1 ijms-23-00723-t001:** Clinical application of decellulized homogeneous pulmonary valves.

Trial (Year)	Type of Homograft	Mean Age (Years)	Main Findings
ESPOIR (2019) [82]	fDPVH	21.3 ± 14.4	Excellent performance with freedom from explantation and reintervention. Better safety and effectiveness than BJV and CH.
Bobylev et al. (2018) [89]	fDPVH	Range 2–38	Superior mid-term results in children and young adults for PVR. fDPVH provides an alternative therapy for young patients who require multiple valve surgeries.
Sarikouch et al. (2015) [81]	fDPVH	15.8 ± 10.21	One-hundred percent freedom from explantation and endocarditis for fDPVH compared with CH and BJV at 10-year follow-up, associated with no increased valvular gradient.
Cebotari et al. (2011) [83]	fDPVH	12.7 ± 6.1	fDPVH showed the lower mean transvalvular gradient and no cusp thickening or aneurysmatic dilatation. Plus, five-year freedom from explantation was 100%. fDPVH also exhibited adaptive growth.
Cebotari et al. (2006) [65]	fDPVH	Age 11 and 13	fDPVH was feasible and safe with potential to remodel and grow (increase in annulus diameter). There was no sign of valve degeneration at 3.5-year follow-up.
Dohmen et al. (2011) [90]	cDPVH	39.6 ± 10.3	Excellent hemodynamic performance for up to 10 years with no evidence of calcification.
Brown et al. (2011) [91]	cDPVH	28.6 ± 16.0	No patients required reoperation and valve function did not deteriorate. Clinical and hemodynamic performance was encouraging and did not differ significantly from CH
Burch et al. (2010) [92]	cDPVH	9.95 ± 7.96	There was no significant difference in the trend of lower peak valve gradient and re-intervention between cDVPH and CH.
Dohmen et al. (2007) [93]	cDPVH	44.0 ± 13.7	cDVPH showed excellent hemodynamic performance, and may prevent valve degeneration and improve valve durability
Hawkins et al. (2003) [94]	cDPVH	8.5 ± 7.9	After 1 year, the hemodynamic function of cDPVH was similar to that of CH, but the levels of class I and class II HLA antibodies were significantly lower in cDPVH than in CH.

BJV: bovine jugular vein; cDPVH: cryopreserved decellularized pulmonary valve homograft; CH: cryopreserved homograft; fDPVH: fresh decellularized pulmonary valve homograft.

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
