# Peer review of "Tissue Engineered Transcatheter Pulmonary Valved Stent Implantation: Current State and Future Prospect"

_ijms, 2022, doi:10.3390/ijms23020723_

Round 1
Reviewer 1 Report
The article "Tissue-engineered transcatheter pulmonary valved stent im-2 plantation: current state and future prospect" reviews the current state of transcatheter pulmonary valve replacement and the future prospects. The article is well-rounded and can be considered for publication after minor revision.
Abstract: There can be two more sentences added to get continuity. Authors can include the favorable results from transcatheter pulmonary valve replacement to make the abstract more robust. (at line 19 or 21).
Authors say "TPVR currently has a high success rate with favorable safety and prognosis" (line 68). It would be good to quantify this from literature as it helps readers understand the improvement TPVR has gained over the years.
Figures are not referenced properly throughout the manuscript. Figure 1 cannot be seen referenced in the article. Also, the subfigures (fig 1A, 1B, etc) need to be properly referenced.
"A percutaneous pulmonary valved stent designed to reduce the internal diameter of the RVOT has been successfully placed in the RVOT of sheep, but further studies are needed in humans" (line 119-120), what further studies are needed in humans, are the sheep models relevant to human models
Tissue-engineered heart valve (TEHV) needs to be specified when referenced for the first time in the manuscript (line 150 - THEV).
Why are decellularized heart valves more favorable compared to synthetic-material valves for clinical applications? (line-163)
What is meant by adult cells in line-207, are they primary cell lines?
Can the authors include some insight into better cryopreservation techniques used for better TVPR in the article (line-463)
Overall spellings and consistent use of abbreviations need to be checked.
Author Response
Answer to Reviewer 1
Comment 1 Abstract: There can be two more sentences added to get continuity. Authors can include the favorable results from transcatheter pulmonary valve replacement to make the abstract more robust. (at line 19 to 21).
Author’s response: We have now revised accordingly (see line 19-21) or here:
Abstract: “Transcatheter pulmonary valve replacement was used as a non-surgical, less invasive alternative treatment for right ventricular outflow tract dysfunction and had been rapidly developing over the past years.”
Comment 2
Authors say "TPVR currently has a high success rate with favorable safety and prognosis" (line 68). It would be good to quantify this from literature as it helps readers understand the improvement TPVR has gained over the years.
Author’s response:
We have added the following (line 79-82) or here:
“A 96.2% success rate was reported in a previous meta-analysis23. Data from US clinical trial data, the freedom from 5-year reintervention and reinsertion rates were reported to be were 76 ± 4% and 92 ± 3%, respectively24.”
Comment 3: Figures are not referenced properly throughout the manuscript. Figure 1 cannot be seen referenced in the article. Also, the subfigures (fig 1A, 1B, etc) need to be properly referenced.
Author’s response: We are sorry for our incorrect reference and did corrections in line 84, 117, 122, and 130.
Comment 4: "A percutaneous pulmonary valved stent designed to reduce the internal diameter of the RVOT has been successfully placed in the RVOT of sheep, but further studies are needed in humans"(line 119-120), what further studies are needed in humans, are the sheep models relevant to human models?
Author’s response: What we meant to describe was that a percutaneous pulmonary valved stent designed to reduce the internal diameter of the RVOT has been successfully placed in the RVOT of sheep, but it is still in preclinical trials33,34 but further studies are needed in humans (line 132-134).
Comment 5: Tissue-engineered heart valve (TEHV) needs to be specified when referenced for the first time in the manuscript (line 150 - THEV).
Author’s response: We have corrected this problem in the paper, now (line 173-174):
Tissue engineered heart valve (TEHV) THEV
Comment 6: Why are decellularized heart valves more favorable compared to synthetic-material valves for clinical applications? (line-163).
Author’s response: Thank you for your comments, the discussion regarding this question is presented in the following (line 529-537):
“The lack of homografts has prompted researchers to search for new polymeric valve materials. In preclinical studies, polymeric valves have shown promising early function, implantation remodeling, and endothelialization potential115-117. However, they are still not being used in clinical applications. The reasons for this may be manifold; firstly, their long-term safety and efficacy have not been established, yet. Secondly, the whole process, both technically and logistically, is complex. Finally, differences between donor-to-donor may lead to inconsistent and uncontrolled leaflet thickening and shortening of the final product, resulting in valve insufficiency118. Out of these reasons decellularized heart valves are more favorable than synthetic ones at the time81,82.
Comment 7: What is meant by adult cells in line-207, are they primary cell lines?
Author’s response: Yes, thank you! The meaning we want to express for adult cells is primary cells. In the case of endothelial cells, for example, human endothelial cells are cultured and proliferated in vitro and then colonized on decellularized homologous heart valves to construct tissue-engineered valves.
We changed it accordingly in line 230-232:
“Seed cells can generally be divided into adult cells (primary cells) and stem cells (or precursor cells that differentiate in a specific direction).”
Comment 8: Can the authors include some insight into better cryopreservation techniques used for better TVPR in the article (line-463)
Author’s response: Thank you for your suggestion. As suggested by the reviewer we have added the suggested content to the manuscript on line 507-525.
“With the development of multiphoton-induced autofluorescence and second harmonic generated imaging, it is possible to visualize the alterations of the ECM in frozen heart valve tissue109. It is well known that the formation of extracellular ice is a hazard to structural tissues and organs. In 1965, Farrant recommended replacing 60% of cell water with cryoprotectant to prevent freezing at temperatures below -70 °C110. As a result, several valve tissue preservation methods have been developed. To promote vitrification, a 55% cryoprotectant formulation can be used to avoid ice formation below the glass transition temperature of the cryopreservation solution111. Brockbank et al. reported the use of an 83% cryoprotectant formulation. 83% of cryoprotectant formulation are stored at higher temperatures than 55% cryoprotectant formulation and cut without the risk of ice formation. This method is called ice-free cryopreservation. the study was shown that there was no significant difference in material properties, ECM component integrity and cell viability of porcine valve tissue stored under 83% cryoprotectant solution at -80°C compared to -135°C vitrification cryopreservation112. However, cryoprotectants in high concentrations are potentially toxic, so long-term studies are still needed to determine their safety.
While other storage methods such as glutaraldehyde113, glycerol114, etc. have been reported, there is no consensus on the best storage strategies so far to ensure the quality of decellularized TEHV.”
Comment 9: Overall spellings and consistent use of abbreviations need to be checked.
Response: We are very grateful to this reviewer for pointing out this problem. We have now checked the abbreviations in detail, corrected them, and inserted a glossary/abbreviation overview.
Thank you for your careful review. We really appreciate your efforts in reviewing our manuscript. Your careful review has helped to make our review much clearer and more comprehensive.
Reviewer 2 Report
This paper is interesting and valuable for researchers on tissue engineering. However, the authors should introduce the importance of tissue engineering utilizing biomaterial. In addition, the comparison with other studies should be discussed by quoting related papers. Taken together, major revisions should be made before re-submission. The paper would be re-considered only when all the comments were responded.
- Introduction
The authors should explain the specificity of diseases utilizing biomaterial-based tissue engineering. This aspect is essential to understand this research.
I suggest at least these recent review or research papers be added.
(e.g., Biomaterial-based tissue engineering has been recently noted because it can enhance the cell viability or differentiation [1], such as bone [2], skin [3], cancer [4], cardiac [5], or muscle [6].)
[1] For overall concept Int. J. Mol. Sci. 2021, 22(16), 8657
[2] For bone tissue engineering Nanomaterials 2020, 10(8), 1511;
[3] For skin tissue engineering Advanced Drug Delivery Reviews 128 (2011) 352–366
[4] For cancer tissue engineering Cancers 2020, 12(10), 2754
[5] For cardiac tissue engineering Macromol. Biosci. 2018, 18, 1800079
[6] For muscle tissue engineering Regenerative Therapy 18 (2021) 372-383
2.
Despite the review, I think the description or references are poor.
How about cytotoxicity?
Author Response
Reviewer 2
We are very grateful for your comments for the manuscript. According to your advice, we amended the relevant part in the manuscript. All of your questions were answered one by one:
Comment 1: Introduction
The authors should explain the specificity of diseases utilizing biomaterial-based tissue engineering. This aspect is essential to understand this research.
I suggest at least these recent review or research papers be added.
(e.g., Biomaterial-based tissue engineering has been recently noted because it can enhance the cell viability or differentiation [1], such as bone [2], skin [3], cancer [4], cardiac [5], or muscle [6].)
[1] For overall concept Int. J. Mol. Sci. 2021, 22(16), 8657
[2] For bone tissue engineering Nanomaterials 2020, 10(8), 1511;
[3] For skin tissue engineering Advanced Drug Delivery Reviews 128 (2011) 352–366
[4] For cancer tissue engineering Cancers 2020, 12(10), 2754
[5] For cardiac tissue engineering Macromol. Biosci. 2018, 18, 1800079
[6] For muscle tissue engineering Regenerative Therapy 18 (2021) 372-383
Response: Thank you for your careful review. Tissue engineered heart valves (TEHV) aim to overcome the drawbacks of current valve grafts by creating living, non-cytotoxic, mechanically similar heart valve replacements that can grow and remodel with the patient. Biomaterials play an important role in the study of tissue-engineered heart valves. We have discussed this question in the Introduction section and in the main text and we cited these articles mentioned above in the Introduction section, now: line 65-72, 419-423, 435-438.
Introduction Section:
“Tissue engineering is a multidisciplinary field that aims to develop biological tissues that can be used in the clinical treatment of diseases. Tissue-engineered products have proven to be effective in different applications, such as burn treatment or drug screening. The success of this approach in other medical fields has established the foundation for its application in heart valves17-22. A new alternative therapy to replace faulty valve grafts is emerging – tissue engineered heart valves. These limitations are being overcome by tissue engineered heart valves that are living, non-cytotoxic, mechanically analogous heart valve replacements which can grow and remodel with the patient.”
Main Text:
“Currently, two types of materials are being studied, biological-based materials and synthetic-based materials. Decellularized valves made from collagen, elastin, fibrin, sodium alginate, chitosan, and other biomaterials have advantages in cell adhesion, migration, proliferation, and differentiation. However, their mechanical properties are fragile.”
and
“Combining synthetic scaffolds with biological scaffolds is also being explored, for example, chitosan-modified polycaprolactone porous scaffolds can improve fibroblast adhesion on TEHV99. The use of synthetic heart valves in clinical applications, however, has not yet been reported.”
Comment 2: Despite the review, I think the description or references are poor.
Response: Thank you for your comments. We have intensified the descriptions and added meaningful citations as appropriate: line 145-155, 157-159.
Comment 3: How about cytotoxicity?
Response: Thank you for your precious comments and advice: it is now added in the main text (line 400-413):
“In addition, we should consider the cytotoxicity of decellularized tissue-engineered valves. Detergents commonly used are sodium deoxycholate (SDC), sodium dodecyl sulfate (SDS), and ethylenediaminetetraacetic acid (EDTA). Even after rinsing, detergent residue will remain. After surface seeding, Rieder et al. found that SDS-decellularized xenomatrices had a toxic effect on endothelial and myofibroblast cells46. Nevertheless, residue concentration (<50 mg/L) in the wash solution has not affected the receptiveness of decellularized matrices to human endothelial cells. Therefore, detergent concentrations less than 50 mg/L in the washing solution may be used as an indicator for the production of non-toxic tissues. Alternatively, a combination of detergents at medium concentrations can achieve good decellularization and can also be rinsed off more easily88. Decellularized TEHV needs to have long-term storability. Long-term storage may result in increased decellularized residues due to degradation of the ECM.
The goal of future research should be to find optimal decellularization protocols and storage strategies since decellularization protocols and storage strategies still differ.”
Thank you for your careful review. We really appreciate your efforts in reviewing our manuscript. Your careful review has helped to make our review clearer and more comprehensive. Thanks again!
Round 2
Reviewer 2 Report
The authors have responded to all the comments.
I recommend the publication.